# Association between Maternal Postpartum Depression, Stress, Optimism, and Breastfeeding Pattern in the First Six Months

**DOI:** 10.3390/ijerph17197153

**Published:** 2020-09-30

**Authors:** Andrea Gila-Díaz, Gloria Herranz Carrillo, Ángel Luis López de Pablo, Silvia M. Arribas, David Ramiro-Cortijo

**Affiliations:** 1Department of Physiology, Faculty of Medicine, Universidad Autónoma de Madrid, C/ Arzobispo Morcillo 2, 28029 Madrid, Spain; andrea.gila@uam.es (A.G.-D.); angel.lopezdepablo@uam.es (Á.L.L.d.P.); silvia.arribas@uam.es (S.M.A.); 2Division of Neonatology, Hospital Clínico San Carlos, Instituto de Investigación Sanitaria del Hospital Clínico San Carlos (IdISSC), C/ Profesor Martín Lagos s/n, 28040 Madrid, Spain; gherranz@gmail.com; 3Division of Gastroenterology, Beth Israel Deaconess Medical Center, Harvard Medical School, 330 Brookline avenue, Boston, MA 02215, USA

**Keywords:** breastfeeding adherence, dispositional optimism, exclusive breastfeeding, perceived stress, postpartum depression

## Abstract

Early breastfeeding cessation is a major public health problem. Several factors can affect breastfeeding pattern, and psychological aspects have been poorly explored. We hypothesize that psychological factors and breastfeeding pattern have a relationship. We have assessed in mothers during the first six months of lactation if breastfeeding pattern is associated with maternal stress, postpartum depression, and dispositional optimism, and if these psychological factors play a role on breastfeeding adherence. In total, 711 women participated, answering online the following questionnaires: sociodemographic, perceived stress scale, Edinburgh postpartum depression scale, life orientation test, and breastfeeding adherence score. Women were categorized according to infant feeding practices as exclusive breastfeeding (EBF) or mixed breastfeeding (MBF). The EBF group had a lower score of perceived stress compared to those giving MBF (first month: EBF = 1.5 [1.1; 1.9], MBF = 1.8 [1.5; 2.0]; *p*-Value = 0.030; third month: EBF = 1.6 [1.2; 2.0], MBF = 1.8 [1.5; 2.4]; *p*-Value = 0.038) and also had a lower score of postpartum depression (third month: EBF = 8.0 [6.0; 11.0], MBF = 11.0 [9.0; 15.0]; *p*-Value = 0.001). The breastfeeding adherence score showed a positive correlation with maternal perceived stress (first month: ρ = 0.27; *p*-Value = 0.018), and postpartum depression (third month: ρ = 0.30; *p*-Value < 0.001), and a negative correlation with maternal dispositional optimism (second month: ρ = −0.20; *p*-Value = 0.028). MBF was positively associated with breastfeeding adherence score (odd ratio (OR) = 1.4 [1.2–1.6]; *p*-Value < 0.001) and with postpartum depression (OR = 1.1 [1.0; 1.1]; *p*-Value = 0.020). In the third month of breastfeeding, women with MBF exhibited higher perceive stress and postpartum depression compared to those with EBF and no difference in dispositional optimism. The maternal psychological aspects are associated with breastfeeding pattern. Evaluation of maternal psychological concerns and providing support to lactating mothers may help improving breastfeeding adherence.

## 1. Introduction

Breastfeeding has widely recognized benefits for the infant, the mother, and even public health. The World Health Organization (WHO) strongly recommends exclusive breastfeeding (EBF) which is defined as infant feeding only on breastmilk, without any additional food except vitamins and minerals, until six months of age [1]. Epidemiological studies have shown that breastfeeding cessation has a deleterious impact on infant health in the short [2,3] and long-term [4,5,6]. Black et al. reported 15-times higher relative risk of infant death in children with non-EBF compared to infants on EBF [7]. Besides, breastfeeding cessation can also negatively influence women’s health, both physically [8] and psychologically [9]. Therefore, breastfeeding promotion would be the best cost-effective public health intervention to reduce infant morbidity and mortality, as well as to promote postpartum maternal care [10]. However, worldwide breastfeeding rates are lower than recommended and they decline during the first six months postpartum [11]. In the first month, EBF rates range between 68 and 84%, and by month six postpartum, only 13–20% of women maintain EBF [12,13,14], the European Region exhibiting the lowest rates [15,16,17]. In Spain, only 66% of infants are exclusively breastfed in the first month and the rate drops to 28% in the sixth month [18]. The challenge is to identify the mothers who are at risk of early breastfeeding cessation who may need additional support. A breastfeeding adherence score has recently been designed and widely validated in the Danish population [19], evidencing a strong prediction of breastfeeding cessation in the first four months postpartum, when the drop in EBF rates is higher. The short duration of maternity leave is one of the key factors in breastfeeding cessation [18]; it has been demonstrated that the largest drop occurs in parallel with the return of women to work [11]. However, other factors have also been reported. Among them are early hospital discharge, low breastfeeding support, and receiving advice on formula feeding [20], maternal perception of having an insufficient amount of milk, infant growth failure or mastitis [11]. In addition, maternal psychological factors have emerged as important aspects concerning breastfeeding. Mothers with poor mental or emotional health are less likely to exclusively breastfeed [21,22]. Besides, breastfeeding has a positive psychological impact on the mother, improving her well-being, increasing her self-efficacy and her interaction with the infant [23,24,25,26,27]. On the other hand, early EBF cessation was associated with an increased risk of postpartum depression [9,28], although this association needs to be further explored.

We hypothesize that maternal perceived stress, postpartum depression and dispositional optimism are associated with the breastfeeding pattern. We aimed to explore these relationships. Gaining knowledge on these factors may help to improve guidance for the development of effective breastfeeding intervention programs.

## 2. Materials and Methods

### 2.1. Study Design and Participants Collection

This cross-sectional study investigated breastfeeding mothers using a single online questionnaire comprising five different sections: general sociodemographic questions, Perceived Stress Scale (PSS), Edinburgh Postpartum Depression Scale (EPDS), Life Orientation Test (LOT), and Breastfeeding Adherence Score (BAS). This study was conducted according to the principles of the Declaration of Helsinki, with the approval of the Ethical Committee of Universidad Autónoma de Madrid (PI 19/393-E). The online questionnaire was preceded by the following information: all data were anonymously collected and no IP addresses were recorded, the participation was voluntary, and women had the possibility to end their questionnaire at any time without saving any of their previous responses.

Participants were recruited mainly through non-profit breastfeeding associations, social networks, and maternity-specific discussion boards. This study was referred as “emotional feelings and breastfeeding” in the advertising materials, and terminology that might incite response to social weaknesses was removed to prevent any recruitment bias. The questionnaire was administered in Spanish using the online tool SurveyMonkey (https://es.surveymonkey.com/) and was completed by 2025 participants.

The inclusion criteria were: internet access, willingness to participate in the study, living in Spain, Spanish language comprehension, and being the mother of an infant aged 0–6 months. Exclusion criteria were: lack of answer or typographical error in childbirth date, prior diagnoses of mood disorders, because it increases the odds of postpartum depression [29], or intake of anti-breastfeeding drugs. Mothers of twin infants meeting these criteria were asked to complete only one questionnaire to avoid duplication. The final number of women matching inclusion criteria was 711 (Figure 1).

### 2.2. Socioeconomical Variables and Breastfeeding Period

The following data were collected from each mother: maternal age (years), nationality categorized by origin (European: Spain, France, Sweden, United Kingdom, Slovakia, Poland, Romania, Netherlands, Serbia, and Estonia; North American: United Stated of America, Mexico, Costa Rica, Dominican Republic, Guatemala, San Salvador, Honduras, and Panama; South American: Argentina, Bolivia, Peru, Brazil, Chile, Colombia, Ecuador, Venezuela, Uruguay and Paraguay, and Asian: China and Singapore), first gestation (primiparous; yes/no), gestational age (weeks of gestation), educational level (middle school, high school or university degree), work situation (studying, working or unemployed), family core type (single or two-parents; parents were defined as: biological parents, law guardians, being irrelevant their gender), number of family core members (number of adults and infants living in the household), economic status (average monthly income per capita in the family core, considering the Spanish national average [30]) categorized as: no-income, <1000 €, 1000–2500 €, 2501–4000 €, and >4000 €, mother’s date of birth and child’s date of birth.

The breastfeeding period (in days) was calculated by subtracting the date of childbirth from the date of response to the questionnaire. The breastfeeding period was categorized as the first month (0–30 days), second month (31–60 days), third month (61–90 days), fourth month (91–120 days), fifth month (121–150 days), and sixth month (151–180 days).

### 2.3. Infant Feeding Practices

The questionnaire asked for current infant feeding (exclusive breast milk, breast milk and formula, or exclusive formula). Feeding practices were then categorized as EBF (exclusive feeding on mother’s own milk) or mixed breastfeeding (MBF; infant who had received predominantly formula along with breast milk or other milk, without complementary foods, according to WHO criteria [31]). Mothers who responded that their infants were exclusive formula fed, were included in the MBF group due to low sample size (*n* = 2).

### 2.4. Psychological and Breastfeeding Adherence Instruments

Perceived Stress Scale (PSS) [32]. The Spanish version was used in the present study, which shown an original reliability of 0.82 [33]. Maternal stress was measured using the 10-item Cohen’s PSS with Likert scale responses (0–4). Questions assess, for example, whether in the last month, the mother has felt nervous and stressed, unable to cope, or confident in their ability to handle personal problems. After reverse scoring for items 4, 5, 7, and 8, a sum score divided by ten was obtained. The PSS score was interpreted as the higher the score, the greater the stress. The PSS is not a diagnostic instrument, and therefore has no predefined cut-off values [34]. Cronbach’s alpha for this scale was 0.85.

Edinburgh Postpartum Depression Scale (EPDS) [35]. The present study used the validated Spanish version scale of EPDS [36,37], with an original sensitivity of 79.0% and specificity of 95.5%. The EPDS is a self-reporting scale comprising ten items with the Likert scale response (0–3), including mother’s feelings of anxiety, fear, stress, and enjoyment. Total scores range from 0 to 30 with higher scores indicating higher severity of depressive symptoms. The optimal cut-point of the EPDS was established in ≥ 11 points, with a sensitivity of 79.2% and a specificity of 94.4% [38]. Mothers were asked to fill in the survey according to the feelings experienced in the last seven days. Cronbach’s alpha for this scale was 0.87.

Life Orientation Test (LOT) [39]. This study was adapted and validated for the Spanish population [40,41], with reported original reliability coefficient of 0.66. The revised version of LOT assesses overall optimism versus pessimism using the 6-items with Likert scale respond (1–5). Mothers were asked about their ability to see the positive side of situations, the optimistic perspective of future, attitudes towards tasks, or expectations of beneficial events in their lives. After reverse scoring for items 2, 4, and 5, a sum score divided by six was obtained. The LOT score was interpreted as the higher the score, the higher the optimism. Although LOT can be used to measure dispositional optimism in individual diagnostics, it has been widely used in research and epidemiological studies [42]. Cronbach’s alpha for this scale was 0.80.

Breastfeeding Adherence Score (BAS). This instrument identifies women who may be at risk of breastfeeding cessation in the first four months postpartum [19]. BAS is a novel instrument widely validated in the Danish population and consists of four items with binary and Likert scale responses (1–5). The items were related to educational level, previous experience on breastfeeding, and expectations and feeling about breastfeeding. Total scores range from 0 to 12 with higher scores indicating higher risk of breastfeeding cessation. The cut-point of the BAS was established in ≥ 5 points, with a sensitivity of 80.0% and a specificity of 60.0% [19].

### 2.5. Statistical Analysis

Statistical analysis was performed with R software (version 3.6.0, 2018, R Core Team, Vienna; Austria) within R Studio interface using ggpubr, devtools, arsenal, dyplyr, MASS, nlme, car, oddsratio, and ggplot2 packages. Data were expressed as median and interquartile range [Q1; Q3] and relative frequency in quantitative and categorical variables, respectively. Wilcoxon rank sum test was used to test differences in infant feeding practices in quantitative variables. Fisher exact was used to test differences in proportions between infant feeding practices. Rho-Spearman correlation (ρ) was used to test the association between psychological variables and breastfeeding adherence instruments.

To compare the associations between each of the socioeconomic factors, breastfeeding adherence score and psychological factors, logistic regression models were used to estimate adjusted odd ratio (OR) and 95% confidence interval (CI). Models 1–3 analyzed the influence of psychological factors on breastfeeding pattern. Every variable was introduced stepwise simultaneously. Models were adjusted for the socioeconomic factors, which shown significant differences between groups in the previous univariate analysis. Regression model 4 was used to assess whether psychological factors mediate in the association between breastfeeding pattern and breastfeeding adherence score. This model was similarly built to the models 1–3. The magnitude of change in the OR would mean that the addition of all psychological factors in the model could be mediating the interaction between breastfeeding pattern and breastfeeding adherence score. Significance probability was established at *p*-Value < 0.05.

## 3. Results

Recruitment ended with 711 responses. In terms of women’s nationality, 68.6% were from Spain, 16.9% from Mexico, 6.6% from Chile, and 1.3% from Argentina. The rest of the nationalities were represented in lower than 1.0%.

### 3.1. Socioeconomic Factors and Infant Feeding Practices

Socioeconomic factors stratified by infant feeding practices are shown in Table 1. Maternal age, educational level, work situation, family core, and economic status were not different between women with EBF and MBF. However, most of the mothers who responded to the survey were university-educated, employed, and in a stable relationship, highlighting these three aspects in the EBF group. Primiparous mothers had a significantly higher prevalence of MBF practices than the EBF group. Gestational age was significantly higher in EBF practices than in MBF practices. Both groups had full-term newborns between 38–41 weeks of gestation.

Maternal origin influenced breastfeeding practices, with American mothers being more likely to MBF practice.

Regarding the percentage of mothers with EFB pattern along the period of lactation evaluated, we found 79.2% in the first month 77.1% in the third month, and 81.0% in the sixth month. 

### 3.2. Maternal Psychological Factors Related to Infant Feeding Practices

Mothers giving EBF had a significantly lower score of perceived stress at first and third months of lactation compared to those giving MBF (Figure 2A). EBF group also had a significantly lower score of postpartum depression at the third month of breastfeeding, and a tendency to be lower in the third month compared to MBF group (Figure 2B). Maternal optimism scores were not statistically different between groups along the analyzed breastfeeding period (Figure 2C).

### 3.3. Breastfeeding Adherence score (BAS) and Psychological Factors

MBF group scored significantly higher in the breastfeeding adherence score (BAS) than EBF group from the first month up to the fifth month. A higher BAS score means a higher risk of breastfeeding cessation (Figure 3).

The psychological factors analyzed showed statistical correlations between them at all study points, being negative between PSS and LOT (ρ = −0.40; *p*-Value < 0.001), positive between PSS and EPDS (ρ = 0.73; *p*-Value < 0.001) and negative between EPDS and LOT (ρ = −0.45; *p*-Value < 0.001).

In addition, BAS also showed a statistical positive correlation with PSS in the first month of breastfeeding (ρ = 0.27; *p*-Value = 0.018) and with EPDS in the third month of breastfeeding (ρ = 0.30; *p*-Value < 0.001). Furthermore, BAS showed a statistical negative correlation with LOT in the second month of breastfeeding (ρ = −0.20; *p*-Value = 0.028; Table 2).

### 3.4. Breastfeeding Adherence score and Infant Feeding Practices 

Table 3 reports the results of the logistic regression models examining the association between infant feeding practices, BAS and psychological variables, adjusted by socioeconomic factors, during the first three months of lactation. The cut-point was established in the third month because we have demonstrated that the psychological factors were different between infant feeding practices.

Model 1 showed that during the first three months of lactation, MBF scored a 1.4-fold higher [1.2–1.6] in the BAS than EBF, demonstrating a relationship between breastfeeding pattern and adherence score. Model 2 showed that MBF scored higher on perceived stress, but did not reach statistical significance. Model 3 showed that MBF was significantly associated with postpartum depression. In model 4, the perceived stress and postpartum depression were not significant. However, BAS continued being significant without strong change in adjusted OR. These data suggest a relationship between maternal psychological factors (perceived stress and postpartum depression) and breastfeeding pattern. However, these psychological factors did not seen to mediate breastfeeding cessation, i.e., did not associate with BAS.

## 4. Discussion

This study evidences the influence of maternal perceived stress, postpartum depression, and dispositional optimism on EBF adherence. These psychological factors have been scarcely explored during lactation. Taking into account these factors may help to improve guidance for the development of effective breastfeeding intervention programs.

The adherence to the WHO recommendations of EBF during the first six months is different among European countries. According to data from the Organization of Economic Cooperation and Development, the best result for initiation EBF was in Nordic countries, with rates close to 100% after birth, being the highest percentages in Europe along with Denmark, Switzerland, and Croatia [18]. The United Kingdom had a 30% initiation rate of EBF, and the lowest is in Greece with 21% [16]. However, even in countries with high rates of initial EBF, the percentages of EBF decline over time, being low at six months of age; in Nordic countries, the rates decreased below 20% and in the United Kingdom, they go down to <1% at six months postpartum [17]. According to WHO, the global rate in the European region is 25%, far from the WHO target (50% EBF at six months) [18]. Furthermore, in the United States of America, the global rate of EBF to six months was reported to be 13.8% [43], and in Mexico EBF is lower than 15% [44]. One of the reasons for EBF cessation is the problem of conciliating maternity with work and the duration of maternity leave. Extending maternity leave or changes in the organization of work, such as part-time working or work remotely, have demonstrated to encourage the initiation and longer duration of breastfeeding [45]. Besides, these working practices seem to have a positive impact on women’s mental health in older age [46]. We found that 75% of mothers in our population had EBF; this data is higher than previously reported in Spain, 66% in the study of Theurich et al. [18]. However, it is important to emphasize that our data was collected through breastfeeding social networks, which implies mother’s concern about this issue and therefore increases breastfeeding adherence.

Besides the problem of work–family conciliation, a second difference among countries in breastfeeding adherence could be differences in the Health System practices regarding the way of support, protection, and promotion of breastfeeding [18], as well as socioeconomic and cultural factors. One study shows that educating mothers about breastfeeding is more important than the mother’s level of education, for the maintenance of EBF [11]. Another study of Chinese-born women who migrated to Spain and of Spanish native women concluded that educational and socioeconomic levels did not seem to explain the lower rates of EBF in the Chinese-born women [47]. Similar to this study, we found no statistical differences in these socioeconomic variables between EBF and MBF groups. The proposal of this inequity paradigm has been linked to working conditions as well as cultural characteristics, such as their overall attitude towards breastfeeding [48]. Our data showing that maternal origin is one of the most significant variables influencing breastfeeding pattern support this theory.

Most of the mothers who participated in our study have university studies, with medium-high economic level and with family support from their partners, which may influence the high rates of EBF. Besides, they can also be related to the way of participants’ recruitment and dissemination of the study through breastfeeding support associations, specific forums and social networks, mostly followed by mothers who are interested in breastfeeding. Additional studies with different modes of enrollment, i.e. through primary care centers, could provide additional information on the rates of EBF in Spain.

We explored the influence of maternal psychological factors in breastfeeding pattern. We found that MBF was associated with maternal perceived stress and postpartum depression in the first three months of postpartum. This is a critical period of breastfeeding in Spain, since maternity leave lasts sixteen weeks [49]. Therefore, in this third-month mothers have to decide how to combine breastfeeding with work. Therefore, our data suggest that the psychological barrier of the 3 months postpartum to end EBF could be modified if these psychological factors are taken into account, which could help health professionals to detect women at risk of early cessation of EBF.

The score of BAS was also positively correlated with maternal perceived stress in the first and third months, postpartum depression in the third month, and negatively with maternal optimism in the second month of lactation. This score has been designed and widely validated in the Danish population [19], evidencing a strong prediction of breastfeeding cessation in the first four months postpartum. Our data suggest a relationship between stress and postpartum depression and breastfeeding pattern. However, these psychological factors did not associate with BAS. However, we showed an association between the EBF and maternal stress at the third month of lactation, when the Spanish maternity leave is about to end and the work–family conflicts occur in the family core. 

The link between EBF and maternal mental health may derive from psychosocial and biological factors [29]. Evidence suggests that breastfeeding practices have effects on the mother by reducing her anxiety and stress [25,27]. Breastfeeding attenuates neuroendocrine responses to stress and may operate to improve maternal mood [25,26]. It is also possible that only a subset of women with hormone sensitivity are at risk for depressive symptoms related to breastfeeding cessation. Future research may therefore clarify the maternal context with a modulatory effect on breastfeeding styles and maternal mood.

Several studies have reported that the quality of social and family support is linked to healthier neuroendocrine functioning and positive mood [50,51]. It is therefore not surprising that mothers who lack social support find it more difficult to cope with the challenges associated with EBF, as well as the emotional wear and tear associated with guilt and feelings of inadequacy. These feelings are often associated with early cessation of breastfeeding practices, amplifying the links with maternal stress and psychological discomfort. In addition, due to the high social and educational awareness of the proven benefits of breastfeeding, many women feel intense social pressure to breastfeed. Psychological distress is considered when mothers with severe breastfeeding problems report feeling of guilt and loneliness, which in turn leads to a sense of worthlessness and defeat [27,52].

It is essential to carry out different interventions with breastfeeding mothers and their partners both before and after birth, until exclusive breastfeeding is established. In these interventions, parental education about the importance of skin-to-skin contact immediately after birth, maternal rest, communication, and for mothers to feel understood and supported by nursing professionals should be increased. Likewise, health policies are needed to enable parents to comply with recommendations for breastfeeding adherence, such as longer maternity leave or real reconciliation of family and work for both parents, without the full responsibility for breastfeeding falling on the mother [53,54].

It is then crucial to enhance maternal confidence in her own abilities, enabling mothers to gain a greater understanding of the breastfeeding process and the unique characteristics of infant growth [55], especially in those at greater risk of early cessation of breastfeeding according to the BAS scale.

### Study Strengths and Limitations

First, the temporal association between breastfeeding pattern and maternal psychological factors cannot be established due to the cross-sectional design of this study. The questionnaire was answered at a single time point, in the moment of inclusion. Therefore, we could not obtain data on previous breastfeeding pattern or evolution of psychological factors along lactation. It would be interesting to design a longitudinal study where these questions can be adequately addressed.

Second, despite the fact that the study was conducted in a large sample, there is a possible selection bias due to the recruitment of participants through breastfeeding associations and specific breastfeeding forums. In order to avoid this possible bias, in future studies, it would be interesting to recruit participants through other ways, such as primary care centers, which may add a global perspective breastfeeding practices in the Spanish population.

In addition, although the focus of the present study was on maternal processes, it would be interesting to explore how other obstetrical (such as multiple pregnancies) and neonatal (such as birth weight) variables play a role in the maternal psychological process. Furthermore, how this process could be associated with breastfeeding practices. Even though the work proposal was to explore maternal psychological factors and the breastfeeding pattern, it would be interesting to consider mothers whose infants are exclusively formula fed, which in this work were lacking, to compare them with EBF and MBF groups.

## 5. Conclusions

This study reports breastfeeding pattern in the first six months postpartum among women in Spain, including diverse populations from cultural and ethnical point of view. Although EBF rates were higher than other European countries, EBF declines in the third month of lactation, associated with the end of maternity leave. In addition, the third month was associated with high maternal perceived stress and postpartum depression in women whose infants were fed with MBF. Moreover, breastfeeding adherence scores could predict women at risk of breastfeeding cessation up to the fifth month of lactation. This study highlights some maternal psychological factors that influence the recommended breastfeeding duration among mothers in Spain.

Our study evidences that psychological aspects play a role in breastfeeding pattern, besides sociocultural, work, and family life factors. It would be necessary for a multifaceted, effective, and evidence-based effort to increase European breastfeeding rates, including support and evaluation of maternal psychological concerns.

## Figures and Tables

**Figure 1 ijerph-17-07153-f001:**
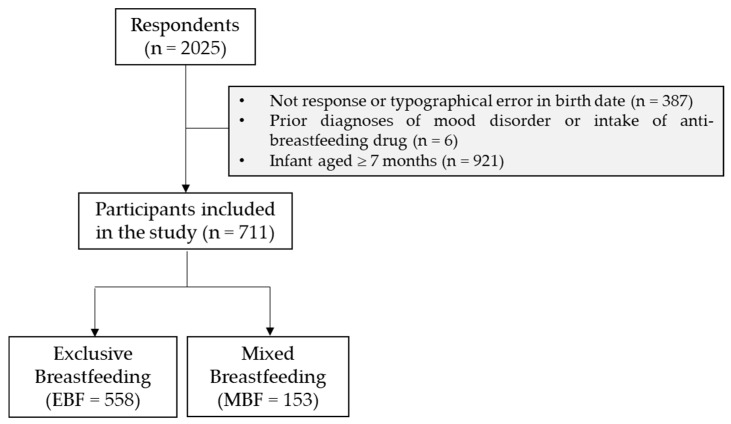
Flow-chart of the responses of the survey and sample size used in the study that match with the inclusion criteria.

**Figure 2 ijerph-17-07153-f002:**
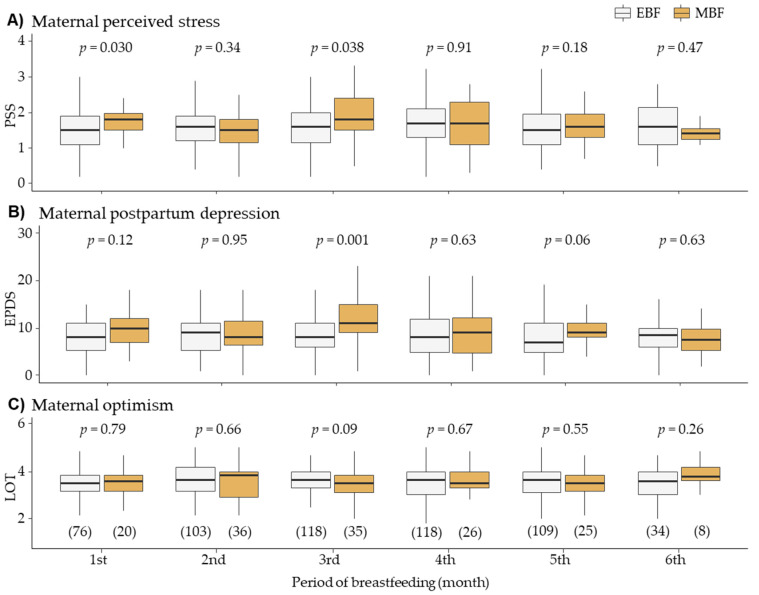
Evolution of psychological factors during the breastfeeding period stratified by infant exclusive breastfeeding (EBF) or mixed breastfeeding (MBF). (**A**) Perceived Stress Scale (PSS). (**B**) Edinburgh Postpartum Depression Scale (EPDS). (**C**) Life Orientation Test (LOT). Data show median and interquartile range. Sample size is shown between brackets. The *p*-Value was obtained by Wilcoxon sum-rank test.

**Figure 3 ijerph-17-07153-f003:**
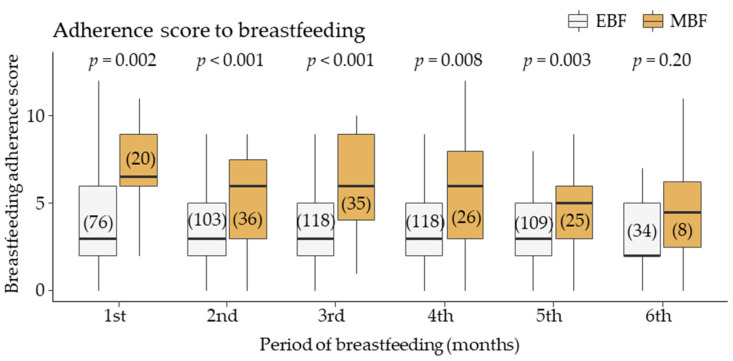
Evolution of breastfeeding adherence score during the breastfeeding period stratified by infant exclusive breastfeeding (EBF) or mixed breastfeeding (MBF). Data show median and interquartile range. Sample size is shown between brackets. The *p*-Value was obtained by Wilcoxon sum-rank test.

**Table 1 ijerph-17-07153-t001:** Socioeconomic factors according to infant feeding practices.

	EBF (*n* = 558)	MBF (*n* = 153)	*p*-Value
Maternal age (years)	33.0 [30.0; 36.0]	33.0 [29.0; 36.0]	0.93
Primiparous	36.7% (205)	43.3% (65)	0.009
Gestational age (weeks)	39.3 [38.3; 40.4]	39.0 [38.0; 39.8]	<0.001
Origin			<0.001
European	74.0% (413)	57.3% (86)	
North American	16.3% (91)	27.3% (41)	
South American	8.2% (46)	12.7% (19)	
Asian	0.2% (1)	0.7% (1)	
**Educational level**			0.65
Middle school	9.9% (55)	11.3% (17)	
High school	17.2% (96)	19.3% (29)	
University degree	72.9% (407)	68.0% (102)	
**Work situation**			0.69
Studying	1.6% (9)	1.3% (2)	
Working	74.9% (418)	72.0% (108)	
Unemployment	23.3% (130)	26.7% (40)	
**Family core**			0.33
Single-parent	12.5% (70)	16.0% (24)	
Two-parents	87.5% (488)	84.0% (126)	
Number of family members	4 [3; 4]	3 [3; 4]	0.11
**Economic status**			0.96
No-income	2.3% (13)	2.7% (4)	
< 1000 €	8.4% (47)	9.3% (14)	
1000–2500 €	45.5% (254)	42.7% (64)	
2501–4000 €	33.3% (186)	35.3% (53)	
> 4000 €	9.0% (50)	9.3% (14)	

Data show median and interquartile range [Q1; Q3] and relative frequency (sample size) in the quantitative and categorical variables, respectively. Exclusive breastfeeding (EBF); Mixed breastfeeding (MBF). Wilcoxon rank sum test for quantitative variables and Fisher exact test for categorical variables. *p*-Value < 0.05 was considered statistically significant.

**Table 2 ijerph-17-07153-t002:** Correlations between breastfeeding adherence score and psychological factors stratified by the breastfeeding period.

Period of Breastfeeding	PSS		EPDS		LOT	
	ρ	*p*-Value	ρ	*p*-Value	ρ	*p*-Value
First month (*n* = 96)	0.27	0.018	0.21	0.05	−0.09	0.43
Second month (*n* = 139)	0.07	0.54	0.02	0.82	−0.20	0.028
Third month (153)	0.13	0.13	0.30	<0.001	−0.15	0.07
Fourth month (*n* = 144)	0.06	0.51	0.10	0.28	0.02	0.81
Fifth month (*n* = 134)	0.14	0.14	0.11	0.31	−0.01	0.97
Sixth month (*n* = 42)	0.09	0.61	0.00	0.81	−0.08	0.70

Data show Rho-Spearman (ρ) correlation and *p*-Value associated. Perceived Stress Scale (PSS); Edinburgh Postpartum Depression Scale (EPDS); Life Orientation Test (LOT).

**Table 3 ijerph-17-07153-t003:** Logistic regression models for the association between breastfeeding practices and maternal perceived stress, postpartum depression, and adherence of breastfeeding during the first 3 months of lactation.

	Model 1		Model 2		Model 3		Model 4	
	OR 95% [CI]	*p*-Value	OR 95% [CI]	*p*-Value	OR 95% [CI]	*p*-Value	OR 95% [CI]	*p*-Value
**BAS**	1.4 [1.2; 1.6]	<0.001	-	-	-	-	1.3 [1.1; 1.6]	0.001
**PSS**	-	-	1.5 [0.9; 2.3]	0.09	-	-	1.0 [0.5; 2.0]	0.95
**EPDS**	-	-	-	-	1.1 [1.0; 1.1]	0.020	1.1 [1.0; 1.2]	0.19

The observation was considered as a mother with ≤ 3 months of breastfeeding (*n* = 449). Exclusive breastfeeding was considered as the reference. Model 1: Breastfeeding Adherence Score (BAS), maternal origin, primiparous rate and gestational age. Model 2: Perceived Stress Score (PSS), maternal origin, primiparous rate and gestational age. Model 3: Edinburgh Postpartum Depression Score (EPDS), maternal origin, primiparous rate and gestational age. Model 4: BAS, PSS, EPDS, maternal origin, primiparous rate, and gestational age. Data show adjusted odd ratio (OR) and 95% confidence interval (CI) with *p*-Value associated.

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
