# Peer review of "Association between Maternal Postpartum Depression, Stress, Optimism, and Breastfeeding Pattern in the First Six Months"

_ijerph, 2020, doi:10.3390/ijerph17197153_

Round 1

Reviewer 1 Report

In the attached file you will find the comments for the authors.

Author Response

This study is relevant due to its subject matter, which is of great interest for the physical and psychological development of babies, contributing from a very early age to the maintenance of positive and sensitive affective relationships between mothers and their children. Therefore, we hope that this work will serve to increase exclusive breastfeeding among mothers and their children in the future. To improve the document, some proposals are made regarding its different sections:

Response: The authors want to thank the reviewer for the time used to review our manuscript and its comments, which we hope have improved the article.

Abstract. Regarding the data, check in the complete document the use of italics in the indexes and the spacing between the = sign, before and after its use, so that it always remains the same. In addition, based on the findings obtained, a final idea must be included to conclude.

Response: We have reviewed all text and used the same type according to the indexes and spacing. In addition, we have included a conclusion in the abstract, which was missing.

Keywords. List keywords in alphabetical order.

Response: the keywords are placed alphabetically in the reviewed version.

Introduction. Line 45: Include a citation of at least one study according to these data.

Response: In table 4 of the reference from Black et al, 2008 published in Lancet, they reported the relative risk of non-exclusively breastfeeding (compared with exclusive breastfeeding from 0 to 5 months). This reference is a special review-series, which Lancet dedicated to mother and child health and their nutrition. 

Method. Regarding the psychological instruments, the reliability indices of the validation study with the original instrument must be indicated in each of the instruments, in addition to the Cronbach's alpha obtained in this study.

Response: Originally reported reliability coefficients of the tests have been added into the main text.

Results. The percentage referring to the nationality of the women in the sample should add up to 100%. These percentages should be reviewed.

Response: The percentages have been revised. However, the nationalities add up to 93.4% because the rest of the nationalities (6.6%) were represented <1.0%.

Lines 236-238: These data should be clarified, indicating which are the direct or indirect relationships, if they exist, and collect it in this way in the discussion.

Response: The data on the association between breastfeeding pattern, BAS (breastfeeding cessation), and psychological variables evidenced that, although there is a relationship between breastfeeding pattern, postpartum depression, and perceived stress (higher in women with MBF), these psychological factors did not seem to mediate the association between breastfeeding pattern and breastfeeding cessation (Model 4 evidenced no relationship). This aspect has now been clarified in the result (lines 279-283) and discussion (lines 349-352).

Discussion. An important part of the references that appear in the discussion are not included in the introduction of the work. Authors should redirect this question, as most of the discussion references should appear earlier in the manuscript introduction. A more descriptive part of the information that appears in the discussion of the work could be advanced to the literature review, being able to cite again in the discussion of the findings when comparing them with the reference studies carried out previously.

Response: We have reviewed the references that all descriptive information is included in the introduction and the data included in them are discussed and compared in the discussion.

Line 260. Are there data referring to the US? It would be interesting to include them because in this study significant differences are obtained in American mothers (in this case Latin American), even though the sample is obtained in Spain (lines 180-181).

Response: We have considered interesting this suggestion and we have added this information and one cite related to USA and another from Mexico (lines 308-310) because there are huge differences between them.

Line 313. The importance of information campaigns both before and after becoming a mother on how to breastfeed children could be added, and in which few cases it would not be indicated, if pertinent. Develop this idea further.

Response: We have added consideration on the importance of carrying out educational interventions by health professionals to parents before and after birth, following them up until exclusive breastfeeding is established, as well as developing health policies that allow real conciliation between family and work of both parents, without placing all the responsibility for breastfeeding on the mother (lines 372-376).  

Conclusions. Line 334. This information does not correspond to or is different from what appears on Line 316. It must be corrected or explained so as not to cause confusion.

Response: In the MBF group (or non-exclusive breastfeeding), we found an increase of PSS and EPDS score at the third month of lactation, compared to the EBF group. However, the LOT score was not significantly different between groups (figure 2). This is the justification for the conclusion sentence and the first part of the discussion sentence. Nevertheless, the PSS and EPDS were significant and negatively correlated with LOT, in both groups, at this time (first paragraph in section 3.3 in the main text). For these reasons, in the discussion we mentioned, “these psychological factors (referred to as PSS and EPDS) were associated with decreased dispositional optimism”.

Line 336. I understand that it is related to the mediation models, but it is not entirely clear. To further explain this result referring to the indirect relationship between the cessation of EBF and maternal psychological factors.

Response: The breastfeeding pattern is associated with BAS and postpartum depression. In addition, BAS can indicate women at risk of breastfeeding cessation. However, the BAS score does not depend on psychological factors. Instead, there is an indirect relationship between breastfeeding patterns and psychological factors. Furthermore, we cannot say that BAS is modulated by maternal psychological variables. Section 3.4 was re-written to increase the readability of the article and we added lines 349-352 in the discussion section.

Reviewer 2 Report

Role of postpartum depression, stress and optimism on breastfeeding patterns in the first six months

The authors searched for an association between postpartum depression, maternal perceived stress, dispositional optimism and breastfeeding.

This is an interesting point. However, we identified several issues considering the methods that the authors need to resolve.

We might be wrong but we understand that the authors pooled together infants who had received mixed breastfeeding and exclusive formula as “mixed breastfeeding”. An interesting point could be to analyze separately mixed breastfeeding and exclusive formula.

The study classified participants in two groups: Exclusive and Mixed breastfeeding. We understand that this is at inclusion. However, do the authors have data on breastfeeding before the inclusion? An interesting point could be to analyze possible breastfeeding modification in the mixed breastfeeding group.

Considering infant’s data, do the authors have some other variables that could be included in the model?

Another bias is that the recruitment of participants through breastfeeding associations and specific breastfeeding forums as mentioned in the limitation section. So, we think the results are not reflecting of the population.

Title: The term of role is confusing. The authors reported an association.

Abstract:

We have assessed in mothers during the first six months of lactation if breastfeeding pattern influences maternal stress, postpartum depression and dispositional optimism, and if these psychological factors play a role on breastfeeding adherence. We think the authors search for an association. The term “influences” should be changed.

A conclusion is missing.

Introduction:

The authors wrote: “We hypothesize that maternal perceived stress, postpartum depression and dispositional optimism modulate the breastfeeding patterns. We aimed to explore these relationships”. However, we understand that the participants only fulfilled once the questionnaire. So, the authors rather searched for an association.

Materials and methods

2.1 Study design

Inclusion criteria: being a mother of an infant aged 0-6 months. We understand that the questionnaire is fulfilled once at the inclusion’s time. If so, we think it is difficult to compare data on breastfeeding when including infant with different ages (Table 1). Does infant age differ in the two groups? Do the authors have data on breastfeeding before the inclusion?

Mothers of twin infants completed only one questionnaire. However, how many women had twins? Was the variable “twins” included as a possible confounding variable?

Considering other possible confounding variables, what about the child? Indeed, low birth weight and/or neonatal hospitalization could be also associated with breastfeeding.

2.2 Socioeconomical variables and breastfeeding period: What is the meaning of “previous work”? What is about current work?

2.3 infant feeding practices

If we well understand, the authors included as MBF infant who had received mixed breastfeeding but also exclusive formula. Do the authors have enough data to analyze mixed breastfeed and exclusive formula separately?

2.5 Statistical analysis: The models should be better described.

Figures should be revised.

Table 3: If the authors choose to keep it, a legend explaining the different models is missing.

Author Response

The authors searched for an association between postpartum depression, maternal perceived stress, dispositional optimism and breastfeeding. This is an interesting point. However, we identified several issues considering the methods that the authors need to resolve.

Response: The authors want to thank the reviewer for the time used to review our manuscript and its comments, which we hope have improved the article.

We might be wrong but we understand that the authors pooled together infants who had received mixed breastfeeding and exclusive formula as “mixed breastfeeding”. An interesting point could be to analyze separately mixed breastfeeding and exclusive formula.

Response: we agree with the reviewer's comment, and we want to explore with more details the breastfeeding pattern on the psychosocial process. However and unfortunately, just two mothers responded that they fed their infants with exclusive formula. For this reason, and ruling out the possibility of a statistically enough sample size, we considered categorizing in mixed breastfeeding as mothers whose infant received breast milk and formula and exclusive formula.

The study classified participants in two groups: Exclusive and Mixed breastfeeding. We understand that this is at inclusion. However, do the authors have data on breastfeeding before the inclusion? An interesting point could be to analyze possible breastfeeding modification in the mixed breastfeeding group.

Response: We agree with the commentary of the reviewer. The study was designed to answer the questionnaire on the feeding pattern and psychological factors in the moment of inclusion. Therefore, we do not have data on the previous pattern and if they have changed it.  It would indeed be very informative to include in a future survey a specific question on how was the initiation of feeding patterns and the reason for cessation (if EBF was the initial pattern). We have added a paragraph in the discussion about this aspect for future research (lines 382-385).

Considering infant’s data, do the authors have some other variables that could be included in the model?

Response: With this manuscript, we wanted to approach breastfeeding pattern from a maternal perspective since the special volume, which we would like to participate with the manuscript, is about "Women's Health throughout Life Stages". We agree on the fact that neonatal variables are key in breastfeeding patterns. However, we did not collect additional data regarding neonates to avoid extending the questionnaire and, because we thought it would be a questionnaire to be answered by the mother herself, who sometimes (i.e. after 6 months from birth), might forget certain neonatal data that would have to be treated as missing data in the analysis.

Another bias is that the recruitment of participants through breastfeeding associations and specific breastfeeding forums as mentioned in the limitation section. So, we think the results are not reflecting of the population.

Response: We agree with the reviewer's assessment and therefore, this limitation was clearly exposed as one of the gaps in the work. However, it must also be assumed that the association consulted was freely formed by mothers, who are concerned about breastfeeding and that, although this may be biased, it is a representative part of the current breastfeeding population. In future studies, we could recruit participants through primary care centers, where the parent takes the infants regularly for revisions during the first 6 months and could avoid the bias of lactation associations.

Title: The term of role is confusing. The authors reported an association.

Response: According to the reviewer’s suggestion, we have changed the title of the manuscript for “Association between maternal postpartum depression, stress, optimism and breastfeeding pattern in the first six months”

Abstract: We have assessed in mothers during the first six months of lactation if breastfeeding pattern influences maternal stress, postpartum depression and dispositional optimism, and if these psychological factors play a role on breastfeeding adherence. We think the authors search for an association. The term “influences” should be changed. A conclusion is missing.

Response: we agree with the reviewer’s comment and we have changed the sentence and added a conclusion in the final paragraph.

Introduction: The authors wrote: “We hypothesize that maternal perceived stress, postpartum depression and dispositional optimism modulate the breastfeeding patterns. We aimed to explore these relationships”. However, we understand that the participants only fulfilled once the questionnaire. So, the authors rather searched for an association.

Response: as the previous comment, we also have modified the sentence to adjust with the reviewer’s suggestion.

Materials and methods.

2.1 Study design: Inclusion criteria: being a mother of an infant aged 0-6 months. We understand that the questionnaire is fulfilled once at the inclusion’s time. If so, we think it is difficult to compare data on breastfeeding when including infant with different ages (Table 1). Does infant age differ in the two groups? Do the authors have data on breastfeeding before the inclusion?

Response: the gestational age of the infant was significantly different between groups and, for this reason, and trying to avoid interferences in the association of psychological variables and breastfeeding patterns, we adjusted the models by gestational age, according to the reviewer’s proposal. It would be very interesting to follow up on the mothers at several points and this is within our future perspectives. One of the reasons for the decision to include a single point was the possibility that an extensive questionnaire along time could lead to participants leaving the questionnaire unanswered. This is an interesting point, which we aim to follow in future studies.

Mothers of twin infants completed only one questionnaire. However, how many women had twins? Was the variable “twins” included as a possible confounding variable?

Response: Only 7 of the 711 mothers included in the analysis had a twin pregnancy, and although the type of pregnancy is a variable to be considered in the breastfeeding pattern, due to the small sample size it was not included in the models. However, it would be interesting to consider this proposal since we have evidence that twin pregnancies play a key role in maternal psychological processes. We have now added this appreciation in section 4.1.

Considering other possible confounding variables, what about the child? Indeed, low birth weight and/or neonatal hospitalization could be also associated with breastfeeding.

Response: As we have previously indicated, the focus of this work was the maternal process. However, we completely agree with the reviewer’s comment and it is already known that neonatal variables are key in breastfeeding patterns. Unfortunately, we did not collect these and we have added it as a limitation of the study in section 4.1.

2.2 Socioeconomical variables and breastfeeding period: What is the meaning of “previous work”? What is about current work?

Response: The word "previous" in the text associated with the work situation was removed to avoid confusion.

2.3 infant feeding practices. If we well understand, the authors included as MBF infants who had received mixed breastfeeding but also exclusive formula. Do the authors have enough data to analyze mixed breastfeed and exclusive formula separately?

Response: Just two mothers responded that they fed their infants with exclusive formula. We did not get enough sample size to be considered separately.

2.5 Statistical analysis: The models should be better described.

Response: In the revised version of the manuscript, the description of the models has been re-written.

Figures should be revised.

Response: both figures have been carefully revised.

Table 3: If the authors choose to keep it, a legend explaining the different models is missing.

Response: The table 3 footnote was changed to explain how the models were built.  

Round 2

Reviewer 2 Report

The authors improved their manuscript and clarified several points.

Response: Thank you for these comments, we have added some details in the manuscript.

However, the description of models in the method' section should be improved.

Response: The paragraph of the statistical section was modified and added more details for the models build.

I might be wrong but I did not find in the manuscript the point that only two mothers responded that they fed their infants with exclusive formula. If so, this point should be added in the manuscript and discussed in the limitation section.

Response: We have added this consideration in the Material and Methods and Limitations’ sections of the manuscript.